# Confidence intervals by constrained optimization—An algorithm and software package for practical identifiability analysis in systems biology

Ivan Borisov *⊕, Evgeny Metelkin⊕

INSYSBIO LLC, Moscow, Russia

⊕ These authors contributed equally to this work.
* borisov@insysbio.com

**Data Availability Statement:** All relevant data are within the manuscript.

**Funding:** The authors received no specific funding for this work.

## Abstract

Practical identifiability of Systems Biology models has received a lot of attention in recent scientific research. It addresses the crucial question for models' predictability: how accurately can the models' parameters be recovered from available experimental data. The methods based on profile likelihood are among the most reliable methods of practical identification. However, these methods are often computationally demanding or lead to inaccurate estimations of parameters' confidence intervals. Development of methods, which can accurately produce parameters' confidence intervals in reasonable computational time, is of utmost importance for Systems Biology and QSP modeling.

We propose an algorithm Confidence Intervals by Constraint Optimization (CICO) based on profile likelihood, designed to speed-up confidence intervals estimation and reduce computational cost. The numerical implementation of the algorithm includes settings to control the accuracy of confidence intervals estimates. The algorithm was tested on a number of Systems Biology models, including Taxol treatment model and STAT5 Dimerization model, discussed in the current article.

The CICO algorithm is implemented in a software package freely available in Julia (https://github.com/insysbio/LikelihoodProfiler.jl) and Python (https://github.com/insysbio/LikelihoodProfiler.py).

## Author summary

Differential equations-based models are widely used in Systems Biology and Quantitative Systems Pharmacology and play a significant role in the discovery of new disease-directed drugs. Complexity of models is a trade off from their employment to crucial fields of biology and medicine. These areas of application require large non-linear models with many unknown parameters. How accurately can the parameters of a model be recovered from experimental data? What is the identifiable subset of parameters? Can the model be reduced or reparameterized to become identifiable? All those questions of identifiability

**Competing interests:** The authors have declared that no competing interests exist.

analysis are essential for model's predictability and reliability. That explains why the topic of identifiability of Systems Biology models has received a lot of attention in recent scientific research. However, existing numerical methods of identifiability analysis are computationally demanding or often lead to inaccurate estimations. Development of methods, which can accurately produce parameters' confidence intervals in reasonable computational time, is of utmost importance for Systems Biology and QSP modeling. We propose an algorithm and a software package to test identifiability of Systems Biology models, designed to speed-up confidence intervals estimation and reduce computational cost. The software package was tested on a number of Systems Biology models, including Taxol treatment model and STAT5 Dimerization model, discussed in the current article.

This is a *PLOS Computational Biology* Methods paper.

# Introduction

## Practical and structural identifiability

Reliability and predictability of a kinetic systems biology model depends on how precisely the parameters of the model can be recovered from the given experimental data. Fitting a model to experimental data is not enough to estimate all the parameters unambiguously. Noisy or incomplete experimental data as well as the models structure often result in uncertainty in parameters estimations.

Identifiability analysis is crucial for models verification. It addresses the question to what extent and with what level of certainty can parameters of a model be recovered from the available experimental data. Two branches of identifiability analysis are distinguished [1] often referred to as *structural* identification and *practical* identification. While structural identifiability is the characteristic of a model's structure and does not take into account available experimental data, practical identifiability considers real noisy and incomplete experimental data.

The goal of *structural* approach [2,3] (*prior* identifiability analysis) is to verify model's identifiability by exploring the model's structure independently from the experimental data. A wide range of methods have been proposed for testing structural identifiability. The strengths and weaknesses of those methods have been thoroughly analyzed in scientific literature [1,4].

*Practical* identification (*posterior* identifiability analysis) is a data-based approach. The approach addresses the possibility and the precision of parameters estimation based on available data. It takes into account the measurement noise and data incompleteness. Hence, parameters' values can be recovered only with some level of certainty, typically described by confidence intervals and confidence regions. The authors of the study [5] define practical identifiability on the basis of profile likelihood notion: identifiable parameter is one that has finite profile likelihood-based confidence interval. Accordingly, the non-identifiable parameters' profile likelihood-based confidence interval is infinite.

Even if a model includes only structurally identifiable parameters it doesn't imply their practical identifiability. While structural non-identifiability implies practical non-identifiability, structurally identifiable models often appear to be practically non-identifiable [6].

Profile likelihood is a reliable though computationally demanding approach to test parameters' identifiability in Systems Biology (SB). It helps us understand how the data can be mapped to parameters' values and how accurate the model predictions are.

Following the definitions of [5], in the current study we propose new algorithm for practical identification and confidence intervals estimation. This algorithm is designed to produce confidence intervals in shorter computational time compared to other profile likelihood-based approaches while controlling the accuracy of estimates. It does not require the intermediate points to lie on the likelihood profile, which leads to less likelihood function calls. We also propose an implementation of the algorithm in a free open source package tested on a number of published kinetic models.

## Materials and methods

### A kinetic systems biology model

A kinetic systems biology model can be expressed as an ODE system:

$$\frac{d\boldsymbol{x}(t)}{dt} = \boldsymbol{f}(\boldsymbol{x}(t), \boldsymbol{u}(t), \boldsymbol{p}) \tag{1}$$

The state vector $\boldsymbol{x}(t)$ denotes variables of the model (e.g. concentrations of molecular compounds or other values), $\boldsymbol{u}(t)$–known input or control (e.g. treatment regime), $\boldsymbol{p}$ –parameters of the model and $\boldsymbol{f}$ is defined by rate laws. $\boldsymbol{x}(t)$ variables can be numerically integrated for the time range $(0, t_{end})$ given nominal initial values $\boldsymbol{x_0} = \boldsymbol{x}(0)$ and parameters $\boldsymbol{p}$.

### Parameters evaluation and point estimates

The subset of unknown parameters can be estimated using the experimental dataset by solving the inverse problem. Typically, not all the variables $x$ are directly measured and observables $\hat{y}_i(t)$ denote experimentally accessible quantities. The observables can be defined as function of $\boldsymbol{x}(t)$, set of additional parameters $\boldsymbol{s}$ (observation parameters) and random values usually representing measurement errors. An important case of measurement error is additive error with known variance:

$$\hat{y}_i(t) = g_i(\boldsymbol{x}(t), \boldsymbol{s}) + \varepsilon_i(t), i = 1 \dots n \tag{2}$$

where $\varepsilon_i$ are the measurement errors and $g_i$ are observation functions, $n$ is the number of measured components.

The unknown parameters $\boldsymbol{\theta} \subseteq \{\boldsymbol{p}, \boldsymbol{s}, \boldsymbol{x_0}\}$ can be estimated by fitting simulated values $y_i(\boldsymbol{\theta}, t) = g_i(\boldsymbol{x}(t, \boldsymbol{p}, \boldsymbol{x_0}), \boldsymbol{s})$ to experimental data $\hat{y}_i$. Assuming the joint distribution of the measurement noise $\varepsilon$ is known, the estimates of parameters $\hat{\boldsymbol{\theta}}$ are typically obtained with MLE approach [7]. It implies maximizing the probability of obtaining $\hat{y}_i$ values, given the model with $\boldsymbol{\theta}$ parameters. This is usually performed by minimizing the corresponding negative logarithm of the likelihood function (objective function):

$$\hat{\boldsymbol{\theta}} = arg \min_{\theta}[l(\boldsymbol{\theta})] \tag{3}$$

$$l(\boldsymbol{\theta}) = -2\log[\Lambda(\boldsymbol{\theta})]$$

The exact choice of the likelihood function $\Lambda(\boldsymbol{\theta})$ is based on measurement error model. For additive error with known variance according to (2) it can be represented as sum of squared residuals:

$$l(\boldsymbol{\theta}) = \sum_{i=1}^{n} \sum_{j=1}^{k} \left( \frac{\hat{y}_{ij} - y_i(\boldsymbol{\theta}, t_j)}{\hat{\sigma}_{ij}} \right)^2 \tag{4}$$

Here the double summation is performed over $n$ –the number of measured components and $k$ –the number of measured time points. $\hat{y}_{ij}$ denote experimental data points, $y_i(\boldsymbol{\theta}, t_j)$–simulated values and $\hat{\sigma}_{ij}^2$ is the error variance.

MLE provides the point estimates $\hat{\boldsymbol{\theta}}$ for the unknown parameters $\boldsymbol{\theta}$ but does not tell us anything about the uncertainty in $\boldsymbol{\theta}$ estimates. Indeed, the estimated parameters $\hat{\boldsymbol{\theta}}$ may not be unique: another set of parameters may give the same objective function value or be very close to it. The accuracy of the estimates can be expressed by confidence intervals or confidence bands.

## Profile Likelihood based confidence intervals

Confidence interval (CI) is an estimate of the unknown parameter which characterizes it by the range of values for particular confidence level $\alpha$. The confidence interval is a better alternative to the point estimate because it gives more information about possible parameter values.

A confidence interval with confidence level $\alpha$ for the parameter $\theta_i$ is an interval defined by probability $P_{\theta_i}(\theta_i^L \leq \theta_i \leq \theta_i^U) = \alpha$. It is important to note that the definition uses the probability term. It implies constructing a confidence interval many times using numerous data samples, which is typically impossible. Researchers often use different asymptotic methods to estimate confidence intervals, which can produce different estimations [8].

Different methods of CI estimation may lead to different definitions of parameters' identifiability. Profile likelihood is one of the most common and robust ways to construct CIs and state practical identifiability of the estimated parameters [9] based on likelihood-ratio test. It implies constructing likelihood-based CIs by exploring $l(\boldsymbol{\theta})$ as a function of a single parameter $\theta_i$ [10]

$$l_{PL}(\theta_i) = \min_{\theta_{j \neq i}}[l(\boldsymbol{\theta})] \tag{5}$$

Corresponding confidence interval for an estimate $\hat{\theta}_i$ with confidence level $\alpha$ is defined by

$$CI_{\alpha, \theta_i} \equiv [\theta_i^L, \theta_i^U] = \{\theta_i : l_{PL}(\theta_i) - l(\hat{\boldsymbol{\theta}}) \leq \Delta_\alpha\} \tag{6}$$

where $\Delta_\alpha$ is $\alpha$ quantile of the $\chi^2$ distribution if the likelihood ratio test is used, $\hat{\boldsymbol{\theta}}$ is the point estimate of the unknown parameters $\boldsymbol{\theta}$ which corresponds to the minimum of $l(\boldsymbol{\theta})$.

Confidence intervals estimation is the major goal of practical identifiability analysis.

According to [5] "*a parameter estimate $\hat{\theta}_i$ is practically non-identifiable, if the likelihood-based confidence region is infinitely extended in increasing and/or decreasing direction of $\theta_i$, although the likelihood (negative log-likelihood) has a unique minimum for this parameter*".

## Available methods

Two general numerical approaches to construct parameters profiles and PL-based CIs are currently developed and implemented in software packages [11–13]. They can be distinguished as *stepwise optimization-based* approaches and *integration-based* approaches. These approaches sequentially calculate $l_{PL}(\theta_i)$ until the profile function reaches the threshold $l(\hat{\boldsymbol{\theta}}) + \Delta_\alpha$.

*Stepwise optimization-based* approaches are based on the definition of $l_{PL}(\theta_i)$. They imply exploring the shape of $l_{PL}(\theta_i)$ by making small steps from the minima $\theta_i = \hat{\theta}_i$ in the increasing or decreasing direction and re-optimizing $l(\boldsymbol{\theta})$ for all $\theta_{j \neq i}$ at each step of $\theta_i$. The smaller $\theta_i$ steps

the numerical algorithm takes while exploring $l_{PL}(\theta_i)$ the more accurate the profiles are. At the same time, re-optimizing $l(\boldsymbol{\theta})$ at each $\theta_i$ step may require thousands of likelihood function calls, which can be inacceptable for high dimensional ODE models. Progressive derivative-based [5] and linearly extrapolated stepping [11] have been proposed to make appropriate steps and more accurate profile estimations.

*Integration-based* approaches suggest obtaining $\theta_i$ profile as a solution of the ODE system. The ODE system itself is derived from optimal conditions for constrained optimization of $l(\boldsymbol{\theta})$ defined in Lagrangian form. Potentially solving the modified ODE system should produce $arg \min_{\theta_{j \neq i}}[l(\boldsymbol{\theta})]$. However, numerical integration of these ODEs requires Hessian of the likelihood function, which is hard or impossible to compute in many real cases. A number of ideas have been proposed to relax the requirements and either approximate Hessian [14] or obtain it from adjoint sensitivity analysis [12].

Various numerical implementations of *stepwise optimization-based* and *integration-based* approaches have been developed [13,15] CI endpoints can be obtained with these methods as sequence of optimizations or numerical integration steps, which is often unstable or computationally expensive. The success of these methods critically depends on the initial step choice, and calculations become even more expensive when parameter is not identifiable or has wider confidence interval than expected. Existing PL methods are mainly focused on visualizing the profiles and stating if the parameter is identifiable or non-identifiable. The accuracy of CI endpoints estimation is in general beyond the scope of these methods.

## Results

### Algorithm

The current study presents a new approach for confidence intervals estimation and profile likelihood-based analysis of identifiability: Confidence Intervals estimated by Constrained Optimization (CICO). It addresses the above-mentioned difficulties of stepwise optimization-based and integration-based PL implementations, namely computational effort, accuracy of CI endpoints estimation and algorithm termination criteria. The key idea of the method is to obtain CI endpoints and avoid the calculation of profiles as the most computationally expensive part of the analysis.

### Method rationale

According to [10] for a given significance level $\alpha$ $CI_{\alpha, \theta_i}$ endpoint values $\theta_i^* = \{\theta_i^L, \theta_i^U\}$ can be found as solutions of the system of $m$ equations:

$$\begin{bmatrix} l(\boldsymbol{\theta}) - l_\alpha^* \\ \dfrac{\partial l}{\partial \theta_j}(\boldsymbol{\theta}) \end{bmatrix} = 0 \tag{7}$$

where $j = 1,\ldots,i-1,i+1,\ldots,m$; $m$ is the number of parameters, and $l_\alpha^* = l(\hat{\boldsymbol{\theta}}) + \Delta_\alpha$ in terms of (6).

Modified version of Newton-Raphson algorithm is proposed in [10] to solve (7) and obtain $\theta_i^*$. Here we propose a different approach to solve (7) based on constrained optimization.

Assuming there exists a solution of (7) and $l(\boldsymbol{\theta})$ possesses derivatives at $\boldsymbol{\theta}^*$, we can denote $\frac{\partial l}{\partial \theta_i}(\boldsymbol{\theta}^*) = s$.

1. In case $s<0$, we can multiply the right and left side of Eq (7) by a positive parameter $\mu = -\frac{1}{s} > 0$ and rewrite the system in the following form:

$$\begin{cases} \mu(l(\boldsymbol{\theta}) - l^*_\alpha) = 0 \\ \mu\frac{\partial}{\partial\theta_i}l(\boldsymbol{\theta}) = -1 \\ \mu\frac{\partial}{\partial\theta_{j\neq i}}l(\boldsymbol{\theta}) = 0 \end{cases} \Leftrightarrow \begin{bmatrix} \mu(l(\boldsymbol{\theta}) - l^*_\alpha) \\ 1 + \mu\frac{\partial}{\partial\theta_i}(l(\boldsymbol{\theta}) - l^*_\alpha) \\ 0 + \mu\frac{\partial}{\partial\theta_{j\neq i}}(l(\boldsymbol{\theta}) - l^*_\alpha) \end{bmatrix} = 0$$

or using matrix notation:

$$\begin{bmatrix} \mu(l(\boldsymbol{\theta}) - l^*_\alpha) \\ \nabla(\boldsymbol{c}^T\boldsymbol{\theta}) + \mu\nabla(l(\boldsymbol{\theta}) - l^*_\alpha) \end{bmatrix} = 0 \qquad (8)$$

Note, that $\boldsymbol{c}^T\boldsymbol{\theta}$ is a hyperplane with normal vector $\boldsymbol{c}^T : c^T_j = \begin{cases} 0, j \neq i \\ 1, j = i \end{cases}$.

The system (8) states the necessary optimality conditions (Karush-Kuhn-Tucker conditions) at $\boldsymbol{\theta}^*$ for the following Lagrangian function:

$$L(\boldsymbol{\theta}, \mu) = \theta_i + \mu(l(\boldsymbol{\theta}) - l^*_\alpha), \qquad (9A)$$

which refers to minimization of target function $f(\boldsymbol{\theta}) = \boldsymbol{c}^T\boldsymbol{\theta} = \theta_i$ with inequality constraint $l(\boldsymbol{\theta}) - l^*_\alpha \leq 0$. The minimal $\theta_i$ value is the lower CI endpoint $\theta^L_i$.

2. Likewise, in case $s>0$ we can denote $\mu = \frac{1}{s} > 0$ and apply the similar transformations to the system (7) to obtain optimality conditions for Lagrangian function:

$$L(\boldsymbol{\theta}, \mu) = -\theta_i + \mu(l(\boldsymbol{\theta}) - l^*_\alpha), \qquad (9B)$$

which refers to minimization of target function $f(\boldsymbol{\theta}) = -\theta_i$ with inequality constraint $l(\boldsymbol{\theta}) - l^*_\alpha \leq 0$ and. The maximal $\theta_i$ value is the upper CI endpoint $\theta^U_i$.

3. $\frac{\partial l}{\partial\theta_i}(\boldsymbol{\theta}^*) = s = 0$ is a special case. In this case $\nabla l(\boldsymbol{\theta}^*) = 0$ and $\boldsymbol{\theta}^*$ is a stationary point of $l(\boldsymbol{\theta})$ which can be a solution of (7) but does not satisfy (8). Theoretically, the CICO algorithm excludes this case and additional assumption $\frac{\partial l}{\partial\theta_i}(\boldsymbol{\theta}^*) \neq 0$ should be made for (7) and (8) to be equivalent. In practice, exact equality $\frac{\partial l}{\partial\theta_i}(\boldsymbol{\theta}^*) = 0$ can hardly happen and derivatives close to zero can be handled by lowering the tolerance of the chosen optimizer and ODE solver.

## Interpretation

In the previous section we have reformulated the problem of confidence intervals estimation in the terms of constrained optimization. This approach has a clear geometrical interpretation. We are looking for tangent hyperplanes to the confidence region $CR_\alpha = \{\boldsymbol{\theta} : l(\boldsymbol{\theta}) - l^*_\alpha \leq 0\}$, which correspond to the minimal and maximal feasible $\theta_i$. For $\boldsymbol{\theta} \in R^2$ the approach can be illustrated by Fig 1. The contour lines reflect confidence regions for different $l^*_\alpha$ values. (**A**) plot stands for identifiable case and (**B**) for non-identifiable. In identifiable case (**A**) each confidence region is limited. Hence, corresponding confidence intervals $CI_{\alpha,\theta_i}$ have finite endpoints. In non-identifiable case (**B**) confidence intervals for parameter $\theta_1$ is infinite and confidence

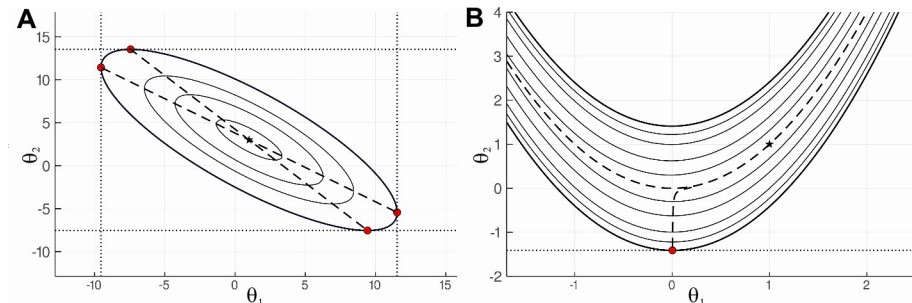

**Fig 1. Contour lines.** Plots show the contour lines of two functions, chosen to illustrate identifiable and non-identifiable cases. Plot (**A**) is an identifiable case illustrated by Booth function $l_A(\boldsymbol{\theta}) = (\theta_1 + 2\theta_2 - 7)^2 + (2\theta_1 + \theta_2 - 5)^2$, which has known minimum $l_A(1,3) = 0$. Plot (**B**) illustrate non-identifiable case by Rosenbrock function $l_B(\boldsymbol{\theta}) = (1 - \theta_1)^2 + 100(\theta_2 - \theta_1^2)^2$ with minimum $l_B(1,1) = 0$. The star-shaped points mark the minima of the above functions. The bold contour represents the $CR_\alpha = \{\boldsymbol{\theta} : l(\boldsymbol{\theta}) - l_\alpha^* \leq 0\}$ for $l_\alpha^* = 200$. The dashed lines are profile paths projected on $(\theta_1, \theta_2)$ Red circles mark the points where tangent hyperplanes correspond to parameters' minimal or maximal values in $CR_\alpha$. Red circles are CI endpoints. The contours were calculated using marching squares algorithm implemented in Contour.jl package (https://github.com/JuliaGeometry/Contour.jl). They are provided for illustrative purposes only.

interval for $\theta_2$ has no finite upper endpoint. CI endpoints were calculated using CICO method.

## Scan bounds and termination criteria

All PL-based approaches: stepwise optimization, integration-based algorithm and CICO imply exploring $\boldsymbol{\theta}$ space by calculating an objective function $l(\boldsymbol{\theta})$ at different $\boldsymbol{\theta}$ points. For a given parameter $\theta_i$ no a-priori information about its identifiability is usually available. In case $\theta_i$ is identifiable we can expect that the profile will intersect with the threshold. In contrast, to state parameter's non-identifiability we have to check all $\theta_i$ feasible values, which can be the whole R space. The definition of practical non-identifiability [9] requires exploration of the whole $\theta_i$ domain but in practice it is never performed. Due to the limitations of computational resources a limited region of $\theta_i$ is often utilized in practice for general identifiability analysis.

To address the discrepancy between identifiability definition and its practical application the numerical implementation of CICO proposes the notion of scan bounds $(\theta_i^{BL}, \theta_i^{BU})$ which represent feasible parameters' values. The scan bounds may be selected based on biologically acceptable values or available computational resources. In practice this approach was utilized by researchers implicitly but the bounds were not used for algorithms termination criteria.

The proposed scan bounds naturally suggest the notion of *practical identifiability within the bounds*. We will call a parameter "*practically identifiable within the bounds*" if its whole confidence interval for a particular confidence level $\alpha$ is located inside the pre-defined scan bounds, i.e. $[\theta_i^L, \theta_i^U] \subseteq (\theta_i^{BL}, \theta_i^{BU})$. If the condition is not satisfied, i.e. $\exists \theta_i^* \in [\theta_i^L, \theta_i^U]$, but $\theta_i^* \in (-\infty, \theta_i^{BL}] \cup [\theta_i^{BU}, +\infty)$ we will call this *parameter practically non-identifiable within the bounds*.

It is necessary to note that the PL-based confidence intervals may be asymmetric relative to $\hat{\boldsymbol{\theta}}$ in contrast to asymptotic confidence intervals. In some cases CIs have finite endpoint in one direction and infinite endpoint in another. In practice it is reasonable to analyze the identifiability of lower and upper sides separately.

The definition of identifiability within the bounds is utilized in the CICO implementation. If lower or upper CI endpoint is present within the scan bounds $(\theta_i^{BL}, \theta_i^{BU})$ the algorithm

converges to the endpoint with preset tolerance. If one of confidence interval's point is found out of scan bounds $(\theta_i^{BL}, \theta_i^{BU})$ the algorithm terminates and the appropriate message is displayed.

## Software implementation: LikelihoodProfiler

We provide an implementation of CICO algorithm in an open source free package Likelihood-Profiler https://github.com/insysbio/LikelihoodProfiler.jl written in Julia language [16]. The package was also translated to free open source package in Python https://github.com/insysbio/LikelihoodProfiler.py. LikelihoodProfiler allows the user to perform CI estimation and state parameter's identifiability. The main function exposed to the end-user is `get_interval` which calculates the upper and lower CI endpoints for the selected parameter $\theta_i$. Currently the CICO implementation depends on NLopt package [17] and the user can choose any suitable optimization algorithm from this package.

To test parameters' identifiability the user should provide `loss_func` which is the likelihood function of unknown parameters $\theta$. The function is expected to be based on MLE approach. The user should also set `theta_init` which is the initial values of parameters which are typically (but not necessary) the optimal values $\hat{\boldsymbol{\theta}}$ obtained by fitting parameters to experimental data. Other mandatory settings are `loss_crit`, which denotes $l_\alpha^* = l(\hat{\boldsymbol{\theta}}) + \Delta_\alpha$ and index denoting the parameter of interest in vector. The user may also set `scan_bounds` which is the feasible $\theta_i$ range $(\theta_i^{BL}, \theta_i^{BU})$, or use the default values (1e-9, 1e9). The following Julia code loads LikelihoodProfiler package and evaluates `theta` endpoints for likelihood function `l(theta)`.

```
using LikelihoodProfiler
l(theta) = 5.0 + (theta[1]-3.0)^2 + (theta[1]-theta[2]-1.0)^2
theta_init = [3.0, 2.0]
ci = [get_interval(theta_init, i, l, loss_crit = 9.0) for i in 1:2]
```

The implementation utilizes two termination criteria, which address two possible situations. In case there is a confidence interval endpoint within the `scan_bounds`, optimization stops when the algorithm converges to the endpoint with the preset tolerance and `BORDER_FOUND_BY_SCAN_TOL` message is displayed. In case the algorithm doesn't find any feasible point above the threshold the algorithm stops with `SCAN_BOUNDS_REACHED` message.

The algorithm can also work in transformed space (`log` or `logit`) which can speed up the optimization process for complex nonlinear models. An optional argument `scale` of `get_interval` function can set search space for each parameter individually. It supports three options:: `direct`,: `log`,: `logit` with default `scale` set to: `direct` for all parameters. The package also includes a set of useful tools for visualization.

Internally LikelihoodProfiler uses Augmented Lagrangian algorithm [18,19] from NLopt package [17], which implies combining the objective function and the constraint into a single function. Then the augmented objective function with no constraints is passed to an optimization algorithm. Augmented Lagrangian implementation used in the package was proved to converge to KKT points [18]. The optimization of the augmented objective function can be performed with any gradient-based or derivative-free algorithm including global optimization methods.

## Validation: The cancer taxol treatment model

Here we provide identifiability analysis of the cancer taxol treatment model [20]. Though the primary goal of this analysis is to verify CI endpoints computed with CICO, we also provide performance estimations of CICO algorithm vs. original implementation [20]. The original

Matlab code is based on stepwise-optimization approach which implies recovering the whole parameters profile to obtain CI endpoint values (https://github.com/marisae/cancer-chemo-identifiability).

The taxol treatment model is defined by the set of ODEs with three state variables, five unknown parameters (a0, ka, r0, d0, kd), dosage regime and experimental data. The unknown parameters have been fitted to experimental data and their estimated values were taken from original Matlab implementation. Even though the model is structurally identifiable, practically available experimental data, as it was shown [20], is insufficient to recover all the unknown parameters.

The same authors provide an open GitHub repository with Matlab implementation of the taxol treatment model (https://github.com/marisae/cancer-chemo-identifiability). This implementation was used to verify the results obtained by CICO algorithm. The repository includes Matlab script for a0 identification. We have adapted this script to estimate CI for other four unknown parameters (ka, r0, d0, kd). No changes were made to the original Matlab code with the exception of counters, which were added to count the number of likelihood function calls the algorithm makes until it reaches the threshold. Internally the Matlab implementation uses lsqcurvefit function for fitting.

To run identifiability analysis with LikelihoodProfiler package the taxol treatment model was rewritten in Julia language. To make the numerical simulations comparable with original Matlab implementation Julia's analogue of Matlab ode23s solver Rosenbrock23 from DifferentialEquations.jl package [21] was used with the same tolerances setup: relative 1e-3, absolute 1e-6. Search bounds for all unknown parameters were set to (1e-3,1e3). CICO CI endpoints were estimated with Nelder-Mead derivative-free solver from NLopt package.

CI endpoints estimated with CICO (**Table 1**) correspond with the values obtained in the original code.

As most of computational efforts in "profiling" approach are focused on solving ODEs with different parameters' sets, the performance of the algorithms was measured by the number of likelihood function calls (**Table 1**) the algorithm makes until it reaches (or converges to) the endpoint. In the taxol treatment model each likelihood function computation requires solving ODE system four times for four different treatment doses.

In general, CICO needs less likelihood function evaluations than stepwise optimization-based profiling to converge to endpoint value. Efficacy of CICO is especially evident in non-identifiable cases. This is due to the constraints incorporated in the objective function as a

**Table 1. Comparison of CICO and stepwise profile likelihood methods for the cancer taxol treatment model.**

| Parameter | LikelihoodProfiler (CICO) | | | | | Original Matlab (Stepwise PL) | | | | |
|---|---|---|---|---|---|---|---|---|---|---|
| | Lower Endpoint | Upper Endpoint | LF Calls (Lower) | LF Calls (Upper) | Time (sec) | Lower Endpoint | Upper Endpoint | LF Calls (Lower) | LF Calls (Upper) | Time (sec) |
| a0 | 6.76 | 17.3 | 285 | 601 | 2.79 | (7.9, 8.32)* | (17.05, 17.46)* | 285 | 1715 | 97.74 |
| ka | 4.99 | 10.73 | 522 | 349 | 3.26 | (4.86, 5.26)* | (10.52, 10.93)* | 682 | 670 | 75.16 |
| r0 | NI | 0.4 | 49 | 796 | 2.85 | NI | (0.36, 0.37)* | 1510 | 7475 | 531.96 |
| d0 | 0.19 | NI | 601 | 170 | 2.81 | (0.13, 0.2)* | NI | 1605 | >20000 | >1000 |
| kd | 50.51 | NI | 796 | 223 | 3.74 | (47.65, 53.61)* | NI | 930 | 12260 | 722.52 |

CI endpoints estimated with CICO and CIs' estimates obtained in the original Matlab stepwise optimization-based implementation. The CI endpoints for original Matlab implementation are given as intervals

(*) because stepwise PL approach doesn't estimate endpoints with any preset tolerance but marks two points before and after parameter's profile intersects the threshold. NI stands for non-identifiable parameter. Elapsed time is measured by @time in Julia and tic toc in Matlab. Computations were performed on a standard desktop computer (2.30 GHz Intel Core i3 with 8 GB RAM).

penalty part. It starts to penalize the algorithm only when optimizer gets near to the threshold, which doesn't happen in many non-identifiable cases where profiles are flat.

Fig 2 illustrates the search path of stepwise "profiling" and CICO for identifiable a0 parameter and non-identifiable kd parameter. Stepwise-optimization tends to follow the profile path while CICO algorithm doesn't require the intermediate points to lie on the profile, which leads to fewer likelihood-function calls.

## Validation: STAT5 dimerization model

STAT5 Dimerization Model [22] consists of eight state variables, nine parameters and experimental dataset. It is proposed as one of the benchmark models in dMod simulation package [13]. We have translated the model from PEtab format used by dMod into Julia. The model's files include best-fit parameter values, which were taken as initial values for identifiability analysis. The boundaries for parameters deviance were set according to PEtab data to (1e-5,1e5). We have reproduced the identifiability analysis of the model in R with dMod and in Julia with LikelihoodProfiler.

dMod implements *integration-based* approach to parameters identification, according to which parameters' profiles are obtained as a solution of ODE system. This approach mentioned in Section 2.4 (Available methods) relies on first derivatives of the likelihood function and Hessian approximation. To ensure the integration accurately follows the profile path each point proposed by integration step can be used as the initial point for optimization. This option is controlled by `method ="optimize"` setting. In case of STAT5 Dimerization Model we have used the"`optimize`" method because default"`integrate`" method had not produced all the profiles due to Hessian-related issues. We have added iteration counter to R code to count likelihood function calls. dMod stops the profile integration when it intersects the threshold or when parameter bounds are reached. Hence, CI endpoints are reported as intervals with average width approximately equal to 3e-2 (Table 2).

This allowed us to set tolerance of endpoint estimation in LikelihoodProfiler `scan_tol = 1e-2.` To make Julia simulations close to deSolve.lsoda used in dMod we have chosen

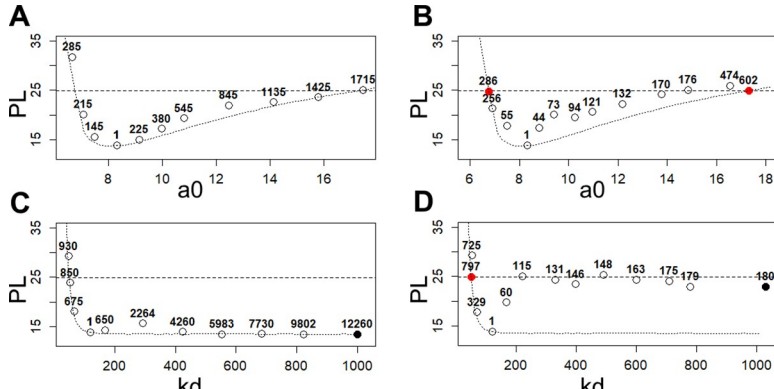

**Fig 2. Search paths for the parameters' CI endpoints of the cancer taxol treatment model.** The path of CI search for stepwise optimization-based algorithm (**A, C**) and CICO algorithm (**B, D**). Circles denote the points reached by the algorithm during the search and numbers above the circles indicate the number of likelihood function calls the algorithm makes to reach this point. The dotted line is the likelihood profile calculated separately for illustrative purposes. The dashed horizontal line marks the significance level $\alpha$ = 0.95. Red circles mark the estimated endpoints (if they exist) for CICO algorithm and black–the points, where the algorithm reaches the box constraints. It denotes non-identifiable case. (**A**) Estimation of lower and upper CI endpoints with the stepwise optimization-based method for a0 parameter. (**B**) Estimation of lower and upper CI endpoints with CICO method for a0 parameter. (**C**) Estimation of lower and upper CI endpoints with the stepwise optimization-based method for kd parameter. (**D**) Estimation of lower and upper CI endpoints with CICO method for kd parameter.

**Table 2. Comparison of LikelihoodProfiler and dMod for STAT5 dimerization model.**

| Parameter | LikelihoodProfiler (CICO) | | | | | dMod (optimize) | | | |
|---|---|---|---|---|---|---|---|---|---|
| | Lower Endpoint | Upper Endpoint | LF Calls (Lower) | LF Calls (Upper) | Time (sec) | Lower Endpoint | Upper Endpoint | LF Calls (Total) | Time (sec) |
| Epo_degradation_BaF3 | -1.71 | -1.42 | 523 | 494 | 0.75 | (-1.74, -1.72)* | (-1.42, -1.39)* | 1716 | 42.15 |
| k_exp_hetero | NI | -3.15 | 4 | 1036 | 0.72 | NI | (-3.1, -3.01)* | 533 | 13.53 |
| k_exp_homo | -2.48 | -1.98 | 237 | 289 | 0.4 | (-2.56, -2.52)* | (-1.95, -1.93)* | 1931 | 47.89 |
| k_imp_hetero | -1.86 | -1.69 | 171 | 179 | 0.32 | (-1.91, -1.9)* | (-1.67,-1.66)* | 1435 | 37.58 |
| k_imp_homo | 0.19 | NI | 1287 | 7 | 1.04 | (0.11, 0.18)* | NI | 2675 | 66.35 |
| k_phos | 4.16 | 4.27 | 143 | 168 | 0.21 | (4.1, 4.12)* | (4.29, 4.3)* | 1959 | 50.75 |
| sd_pSTAT5A_rel | 0.44 | 0.77 | 172 | 243 | 0.34 | (0.42, 0.44)* | (0.78, 0.8) | 2165 | 55.58 |
| sd_pSTAT5B_rel | 0.72 | 0.99 | 231 | 186 | 0.34 | (0.66, 0.68) | (0.99, 1.01) | 2062 | 53.50 |
| sd_rSTAT5A_rel | 0.4 | 0.67 | 204 | 929 | 0.83 | (0.35, 0.36) | (0.67, 0.67) | 2062 | 53.49 |

CI endpoints estimated with LikelihoodProfiler (CICO) and CIs' estimates obtained in dMod. Lower and upper CI endpoints for dMod are given as intervals
* marking two points before and after parameter's profile intersects the threshold. NI stands for non-identifiable parameter. Elapsed time is measured by @time in Julia and system.time in R. Computations were performed on a standard desktop computer (2.30 GHz Intel Core i3 with 8 GB RAM).

LSODA differential equations solver (supported by DifferentialEquations.jl) with the same tolerance setup: relative 1e-7, absolute 1e-7. Nelder-Mead derivative-free solver from NLopt package was used to estimate CI endpoints.

Taking into account the difference of the underlying optimizers, the endpoints reported by LikelihoodProfiler correspond to the values obtained in dMod. The performance of each package was measured by the number of likelihood function evaluations and time required to compute CI endpoints. The results indicate the efficiency of CICO, which on average overperforms integration-based approach implemented in dMod even though dMod relies on model's functions compiled to C. Only for *k_exp_hetero* parameter dMod "optimize" method has recorded fewer likelihood function calls. Timings indicate significant practical efficacy of both CICO and Julia language for this task.

The detailed identifiability analysis of the Taxol treatment model and STAT5 dimerization model, the source code as well as other use-case models' identifiability analyses are published on our GitHub repository (https://github.com/insysbio/likelihoodprofiler-cases).

## Discussion

A number of recent studies have demonstrated that profile likelihood-based methods are efficient to analyze identifiability of the parameters reconstructed on the basis of experimental data. In the absence of identifiability analysis one can never be certain how reliable parameters estimations and how accurate the model predictions are. However, practical usage of profile likelihood-based methods has not become a standard routine yet due to a number of challenges.

Indeed, profile likelihood-based methods are computationally demanding. Progressive stepping and other optimizations of the basic profile likelihood approach impose restrictions on the likelihood function (such as the need to calculate gradients) and limits the set of the applicable optimization methods. The CICO algorithm attempts to solve this problem by replacing multiple calculations of the likelihood function with constrained optimization. For each individual parameter only two optimization iterations are required to calculate the lower and upper CI endpoints. CICO doesn't require the gradient of the likelihood function and allows the user to choose derivative-free or gradient-based optimization algorithm.

Other challenges originate from uncertainty in practical non-identifiability definition. It is implied that researchers have to scan sufficiently wide but finite intervals to state a non-identifiable case. In practice it is usually performed by visualizing the profiles on a chosen interval and extrapolating profiles behavior to the global parameters feasible region. In the current study we have proposed a formal criteria of the algorithm termination, utilizing the scan bounds notion, which can automate the analysis process and get rid of subjectivity.

The numerical experiments have demonstrated that confidence intervals obtained with CICO algorithm coincide with the results reported in the publications. As it was shown, on average the algorithm overperforms considered above optimization-based and integration-based PL implementations. This comparison was performed with the default solver settings and can possibly be optimized for greater efficiency. Moreover, the optimization-based PL approach doesn't converge to the endpoint, while the CICO algorithm was developed to accurately estimate CI endpoints. Hence a more thorough comparison of the algorithms is difficult, since the termination criteria of the optimization-based PL doesn't take into account the accuracy of CI endpoints estimation.

To compare the methods we have measured efficacy in terms of elapsed time and likelihood function calls required to obtain CI endpoints. In general, CICO implementation in LikelihoodProfiler is about 100 times faster than dMod integration-based approach (R) and optimization-based method (Matlab). However, it is important to note that timings highly depend on the programming language, optimization method and ODE solver used while the number of likelihood function evaluations is a language independent measurement, though it also is affected by the efficacy of optimization algorithm and ODE solver.

In addition to confidence intervals, other interval estimates may also be of interest: confidence n-dimensional parameters' regions, prediction bands, etc. The CICO algorithm usage can be potentially expanded to calculate these generalizations of confidence intervals, and we plan to test its use for these classes of tasks in our future studies.

## Author Contributions

**Conceptualization:** Evgeny Metelkin.

**Data curation:** Evgeny Metelkin.

**Investigation:** Ivan Borisov.

**Supervision:** Evgeny Metelkin.

**Validation:** Ivan Borisov.

**Visualization:** Ivan Borisov.

**Writing – original draft:** Ivan Borisov.

**Writing – review & editing:** Evgeny Metelkin.

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
