## [Decision Letter · Decision Letter 0]

4 Sep 2020

Dear Mr. Borisov,

Thank you very much for submitting your manuscript "Confidence intervals by constrained optimization – an algorithm and software package for practical identifiability analysis in Systems Biology" for consideration at PLOS Computational Biology. As with all papers reviewed by the journal, your manuscript was reviewed by members of the editorial board and by several independent reviewers. The reviewers appreciated the attention to an important topic. Based on the reviews, we are likely to accept this manuscript for publication, providing that you modify the manuscript according to the review recommendations.

Sincerely,

Daniel A Beard

Deputy Editor

PLOS Computational Biology

Daniel Beard

Deputy Editor

PLOS Computational Biology

[LINK]

Reviewer's Responses to Questions

**Comments to the Authors:**

Reviewer #1: The article is very clearly written. The authors lay out a timeline of the developments of the field that to me seems fairly complete and leads directly to the new method as a substantial increase in the field. The motivation for focusing on the structural identifiability is made fairly clear. The method is well-described and after reading it, it's clear that it should work. The evidence then demonstrates that it does work. I can easily see this method and this package being used by many researchers in practice.

That said, there are some improvements that should probably be made to the paper before publication. For one, I think that section 3.4 is unnecessary. I think it's fairly clear that this kind of numerical method needs to be computed on some finite support so practically all determinations are going to be made in some box. I don't think that more than a sentence or a paragraph is really required to get that point across. Secondly, the paper itself doesn't seem to have a lot of the validation. One example is used as validation, but the paper needs more. When I look at the package they discuss, I can see 5 clear examples with Binder links that demonstrate the method on more systems: some of this should be in the paper instead of 3.4 in order to more broadly demonstrate the validity of this method. Next, what they established was "structural efficiency", i.e. efficiency in terms of likelihood function evaluations. But it would've been nice to also see "practical efficiency", i.e. raw timings for the MATLAB method and Julia and Python implementation of the new methods, and use this to demonstrate a clear orders of magnitude actual performance improvement. Overall I think it's a really good paper, a good idea, and a strong result with just some touch-ups requires to really hammer home the advance in a more clear way.

Reviewer #2: This article presents a novel method to study practical identifiability of parameters of ODE-based models. The method is innovative and seems to overcome existing methods in terms of computational cost, at least in the presented example. It can definitely be useful for the Research community, especially since the authors have made it freely available either in Julia or in Python. The article is very clear and well written. It cites all relevant literature. I have three minor comments:

-Equation 7 : precise the values for j, to make clearer the fact that this is a system of more than 2 equations.

-Equation 8: it is not obvious how the authors transformed system ([Disp-formula pcbi.1008495.e025]) into system ([Disp-formula pcbi.1008495.e031]). More explanations are needed here since this is key to understand the algorithm. Are the systems strictly equivalent? In the definition of c, the authors need to precise the position of the “1” in the vector.

-The authors claim in the Abstract and Introduction that their method provides more accurate estimation of confidence Interval bounds. However, this is not demonstrated in the article, neither theoretically, nor computationally (On the opposite, they do provide some evidence of the lower computational cost of their algorithm compared to existing ones). Please either add the corresponding evidence or modify the text.

**Have all data underlying the figures and results presented in the manuscript been provided?**

Reviewer #1: Yes

Reviewer #2: Yes

PLOS authors have the option to publish the peer review history of their article (what does this mean?). If published, this will include your full peer review and any attached files.

Reviewer #1: **Yes: **Christopher Rackauckas

Reviewer #2: No
---

## [Decision Letter · Decision Letter 1]

6 Nov 2020

Dear Mr. Borisov,

We are pleased to inform you that your manuscript 'Confidence Intervals by Constrained Optimization – an Algorithm and Software Package for Practical Identifiability Analysis in Systems Biology' has been provisionally accepted for publication in PLOS Computational Biology.

Best regards,

Daniel A Beard

Deputy Editor

PLOS Computational Biology

Daniel Beard

Deputy Editor

PLOS Computational Biology

Reviewer's Responses to Questions

**Comments to the Authors:**

Reviewer #1: The authors have addressed my previous concerns and demonstrate a significant improvement to the practical application of practical identifiability analysis with these new results. In addition, I can confirm that their code, timing, and results on the Julia side are easily reproducible.

Reviewer #2: The authors have answered all my comments.

**Have all data underlying the figures and results presented in the manuscript been provided?**

Reviewer #1: Yes

Reviewer #2: Yes

PLOS authors have the option to publish the peer review history of their article (what does this mean?). If published, this will include your full peer review and any attached files.

Reviewer #1: **Yes: **Chris Rackauckas

Reviewer #2: No

---

## [Editor Report · Acceptance letter]

1 Dec 2020

PCOMPBIOL-D-20-01281R1 

Confidence Intervals by Constrained Optimization – an Algorithm and Software Package for Practical Identifiability Analysis in Systems Biology

Dear Dr Borisov,

I am pleased to inform you that your manuscript has been formally accepted for publication in PLOS Computational Biology. Your manuscript is now with our production department and you will be notified of the publication date in due course.

With kind regards,

Nicola Davies
